# S³ADNet: Sequential Anomaly Detection with Pessimistic Contrastive Learning

## Abstract

Anomalies are commonly found in sequential data generated by real-world applications, such as cyberattacks in network traffic, human activity changes in wearable sensors. Thanks to the development of computing technology, many impressive results have been obtained from deep learning-based anomaly detection approaches in recent years. This paper proposes a simple neural network framework for detecting anomalies on sequential data, called *S*elf-*S*upervised *S*equential *A*nomaly *D*etection *N*etwork (S³ADNet). S³ADNet first extracts the representations from each data point by performing feature augmentation for contrastive learning; then captures the contextual information from the sequential data points for estimating anomaly probabilities by optimizing the context-adaptive objective. Here, we design a novel loss function based on a pessimistic policy, considering that only anomalies can affect the contextual relationships in sequences. Our proposed method outperformed other state-of-the-art approaches on the benchmark datasets by F1-score with a more straightforward architecture.

## 1 Introduction

Sequential anomaly detection (SAD), a subtask of anomaly detection (AD), focuses on detecting outliers or change points for sequential data. Sequential data are produced at a sub-millisecond rate by multifarious real-world applications, usually being of high dimensionality and high noise. Besides the complex nature of data, the sequential data points can have shifting distributions and relationships in different contexts, resulting in the concept drift problem (Žliobaitė, 2010; Gama et al., 2014). Because anomalies can imply certain occurrences that endanger public safety, life, and property, such as intrusions in network connections (Jyothsna et al., 2011; Samrin & Vasumathi, 2017; Moustafa et al., 2019), activity changes and health irregularities in human sensing (Nweke et al., 2018; 2019; Serhani et al., 2020), abnormal events in video surveillance systems (Popoola & Wang, 2012; Mabrouk & Zagrouba, 2018), and risky signals in space technology (Hundman et al., 2018; Tariq et al., 2019; Yairi et al., 2017; Shin et al., 2020), SAD is a challenging but crucial task (Pang et al., 2021; Ruff et al., 2021). Learning meaningful representations from the data and capturing the contextual relationships is the key to solving SAD tasks.

Generally, AD methods are in the unsupervised scheme because anomalies are considered to be much less than normal instances and hard to model. The approaches that train the models exclusively on the normal samples to identify the anomalies are one-class classification methods, such as one-class support vector machine (OC-SVM) (Schölkopf et al., 1999). Since deep neural networks are expert at extracting representative features from high-dimensional and high-noise data, more and more methods leverage deep learning to handle AD tasks. Especially, autoencoders (AEs) (Hinton & Zemel, 1994; Kingma & Welling, 2013) and generative adversarial networks (GANs) (Goodfellow et al., 2020) are frequently used as they are suitable for unsupervised learning, such as deep autoencoding Gaussian mixture model (DAGMM) (Zong et al., 2018), and AnoGAN (Schlegl et al., 2017). However, these methods could be weak in capturing the contextual information from sequences, so that some other approaches introduce recurrent neural networks (RNNs) (Rumelhart et al., 1986) to learn the sequential relationships. For example, the multi-scale convolutional recurrent encoder-decoder (MSCRED) (Zhang et al., 2019), OmniAnomaly (Su et al., 2019), and the temporal hierarchical one-class network (THON) (Shen et al., 2020) apply RNNs for time-series

data modeling. Nevertheless, most of these methods still require picking out the normal data from the mess to train the one-class classifiers or generative models in practice.

This paper proposes a self-supervised sequential anomaly detection network ($S^3$ADNet) to learn the representations of sequential data points and extract the contextual relationships for anomaly detection. Inspired by the simple framework for contrastive learning of visual representations (SimCLR) (Chen et al., 2020), we make the network generate two similar embeddings for input data points with feature augmentation, and the feature extraction layers capture the essential identities of the data points by minimizing the contrastive loss. By optimizing the contextual contrastive loss based on a pessimistic assumption system, the multi-conceptual context (MCC) layer estimates anomaly probabilities with the sequential relationships modeled by the learned representations for each data point. We can use arbitrary parameter-shared neural networks for the representative feature extraction. Additionally, we employ the MCC layer rather than RNN structures for a more intuitive context learning. The proposed model does not require any label information or even selecting the normal data for training one-class classifiers, yet makes the data points compare with each other to recognize anomalies. In the experiment, we compared our model's estimation results to the state-of-the-art approaches using a widely used network traffic dataset and a human activity sensing dataset. Our proposed $S^3$ADNet obtained competitive F1-score results to those methods with a simpler network architecture.

## 2 RELATED WORK

In this section, we review existing research endeavors related to our work. We first overview the approaches for AD and SAD tasks. Then, we outline recent contrastive learning methods for representation learning.

### 2.1 ANOMALY DETECTION

In a narrow sense, AD indicates outlier detection (OD), where the target is the data points with extraordinary properties from most of the rest. Other than outliers, people are more concerned with change point detection (CPD) on sequential data since the abrupt characteristic shifts can intimate valuable and crucial information for the temporal trends.

OD methods can be grouped into three categories based on the main principle of modeling: 1) boundary-based, 2) density-based, and 3) reconstruction-based approaches. Boundary-based methods use certain distance indexes in the model optimization or the anomaly score calculation. OC-SVM is a typical boundary-based method that learns a kernel-based boundary surrounding normal data points on a hyperplane (Schölkopf et al., 1999). Isolation Forest (IF), another widely used approach, constructs trees by randomly splitting branches throughout randomly selected features and considers that the average feature path length of an anomaly to the root is larger than that of a normal instance (Liu et al., 2008). The main idea of classic density-based methods is performing probability density estimation to maximize the likelihood for normal samples, such as using Gaussian mixture model (GMM) for parametric estimation or kernel density estimation (KDE) for nonparametric estimation (Laxhammar et al., 2009; Kim & Scott, 2012). The online algorithm employed a discounting strategy to update the density for sequences (Yamanishi et al., 2004). With the development of deep learning, deep structured energy-based models (DSEBMs) were proposed to model the distribution of normal data points by using deep architectures (Zhai et al., 2016). Another deep method, DAGMM, applies autoencoder to learning data's representations and predict the likelihood based on GMM (Zong et al., 2018). Reconstruction-based methods aim to well restore data from compressed feature space for normal data points but not anomalies. Principal component analysis (PCA) and its variants are traditional ways for data reconstruction (Hawkins, 1974; Schölkopf et al., 1997; Tharrault et al., 2008), while variational AE is of deep style without handcrafted features for the sake (An & Cho, 2015). Methods using GANs can be considered to be hybrid approaches, for they model the distribution by regenerating data points in the zero-sum-game scheme, such as AnoGAN (Schlegl et al., 2017), adversarially learned anomaly detection (ADLD) (Zenati et al., 2018), Mahalanobis distance-based adversarial network (MDAN) (Hou et al., 2020).

For CPD, there is a similar taxonomy to OD. Fast low-cost online semantic segmentation (FLOSS) calculates the contextual boundaries based on the shapes of sequential patterns to eject change points

(Gharghabi et al., 2019). The autoregressive model updates the parametric probability density function across sequential data points (Yamanishi & Takeuchi, 2002). Relative unconstrained least-squares importance fitting (RuLSIF) identifies whether two contiguous subsequences are from different distributions using the nonparametric divergence estimation (Liu et al., 2013). KL-CPD gives the distribution discrepancy by using a learnable deep kernel function and reconstructing samples from two sequential segments with RNNs (Chang et al., 2019). Entropy and shape aware time-series segmentation (ESPRESSO) is a hybrid approach that combines the temporal density and distance to detect changes and obtained better performance than employing one principle (Deldari et al., 2020). Another self-supervised contrastive learning method TS-CP$^2$ takes the subsequences which are not contiguous with the current subsequence as the negative samples for the training and then predicts whether the two subsequences are from the same distribution (Deldari et al., 2021). TS-CP$^2$ employs temporal convolutional networks (TCNs) (Bai et al., 2018) to learn contextual representations, while ours introduces the MCC layer to capture the contextual information and allows using arbitrary feature extraction networks for each data point.

## 2.2 CONTRASTIVE LEARNING

Recently, contrastive learning (CL) has grabbed much attention due to the benefit of avoiding the high cost of labeling large-scale datasets for deep representation learning in an unsupervised/self-supervised manner (Jaiswal et al., 2021). CL has been used for various data formats, including images (Chen et al., 2020; He et al., 2020), time series (van den Oord et al., 2018; Franceschi et al., 2019), videos (Sermanet et al., 2018; Qian et al., 2021), texts (van den Oord et al., 2018; Gao et al., 2021), and graphs (You et al., 2020; Zhu et al., 2020).

The basic strategy of CL is to train a model that makes the feature distances between similar instances closer but the ones between diverse instances as far as possible. For this sake, CL methods design pretext tasks to optimize the distances and use contrastive loss or triplet loss as the learning objective (Chen et al., 2020). Recently, data augmentation has been commonly introduced into the pretext tasks to help the feature extraction. While image data are relatively easy to augment by using conventional computer vision techniques (Shorten & Khoshgoftaar, 2019), there are several word-level (Wei & Zou, 2019) and sentence-level (Kobayashi, 2018) approaches for textual data augmentation. For other data formats, some domain-agnostic methods were proposed, such as adding Gaussian noise (GN) to embeddings (DeVries & Taylor, 2019) or dropout noise (DN) in representation layers (Gao et al., 2021; Liang et al., 2021). GN and DN are not only easy to implement but also have low computational costs for CL tasks.

In SimCLR, given a cluster of augmentation $\Lambda$ and an input $\boldsymbol{x}$, two augmentation operator $\lambda_a \sim \Lambda$ and $\lambda_b \sim \Lambda$ generate two similar instances $\boldsymbol{x}_a = \lambda_a(\boldsymbol{x})$ and $\boldsymbol{x}_b = \lambda_b(\boldsymbol{x})$. Then, the feature extraction network $f$ encodes $\boldsymbol{x}_a$ and $\boldsymbol{x}_b$ into two representations $\boldsymbol{h}_a = f(\boldsymbol{x}_a)$ and $\boldsymbol{h}_b = f(\boldsymbol{x}_b)$. Next, the head network $g$ projects $\boldsymbol{h}_a$ and $\boldsymbol{h}_b$ into two embeddings $\boldsymbol{z}_a = g(\boldsymbol{h}_a)$ and $\boldsymbol{z}_b = g(\boldsymbol{h}_b)$, whose similarity is maximized by minimizing a contrastive loss, the NT-Xent loss:

$$\ell_{NTX}(a, b) = -\log \frac{\exp\left(\text{sim}\left(\boldsymbol{z}_a, \boldsymbol{z}_b\right)/T\right)}{\sum_{n=1}^{2N} \exp\left(\text{sim}\left(\boldsymbol{z}_a, \boldsymbol{z}_n\right)/T\right)\left[n \neq a\right]} \tag{1}$$

$$\text{sim}(\boldsymbol{u}, \boldsymbol{v}) = \frac{\boldsymbol{u}^\top \boldsymbol{v}}{\|\boldsymbol{u}\|\|\boldsymbol{v}\|}, \tag{2}$$

where $T$ is temperature, and $N$ denotes the minibatch size. Since the value range of Equation 2 is $[-1, 1]$, it is usual to set $0 < T < 1$ for an efficient learning. Here, we can regard the similarity as state energy $E$. According to the Boltzmann distribution, the probability of a state $\rho$ satisfies $\rho \propto \exp\left(-E/kT\right)$, where the right hand side is called the Boltzmann factor, and $k$ is a constant (Kittel, 2004). If we use similarity to represent a sequential relationship, we can model the probability of the relationships. Also, we can adapt the temperature to control the sensitivity of the relationship's strength to the probability. This finding motivated the objective function design in our method, which we will describe in Section 3.2.2.

## 3 METHODOLOGY

Before presenting the details of our proposed method, we depict the overview of S$^3$ADNet's architecture, as shown in Figure 1. In the following sections, we first introduce the assumptions behind

pessimistic contrastive learning (PCL). Then, we illustrate the pretext tasks and objective functions in our method.

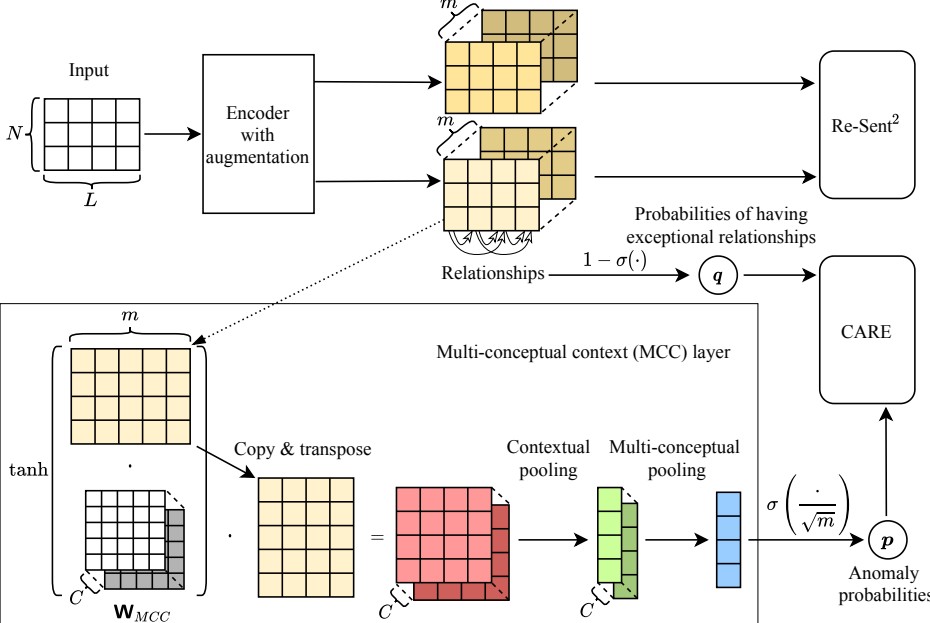

Figure 1: The architecture of S$^3$ADNet. Here, $\sigma\left(\cdot\right)$ indicates the sigmoid function, $\cdot$ symbolizes the tensor dot operation, $C$ signifies the number of concepts, and $\mathbf{W}_{MCC}$ stands for the weight of the MCC layer. The two yellowish tensor blocks are different augmented sequential embeddings generated from the batch input, while the yellow matrix is one sequential embedding for an instance of the calculation in the MCC layer. The exact numbers of cells imply the identical dimension: one of the minibatch size $N = 3$, the sequence length $L = 4$, and the embedding size $m = 5$. The lookahead size for collecting relationships is 2.

### 3.1 ASSUMPTIONS

Given a sequence of elements $[\boldsymbol{z}_1, \boldsymbol{z}_2, \cdots, \boldsymbol{z}_i, \cdots, \boldsymbol{z}_j, \cdots, \boldsymbol{z}_{L-1}, \boldsymbol{z}_L]\,(1 \leq i < j \leq L)$, we model the anomalies based on the following pessimistic policy.

**Assumption 1 (existence of anomalies)** *Anomalies can occur at any element in the sequence.*

We assume that there are anomalies in the training data. Picking out normal samples from a large-scale high-dimensional dataset is expensive, not to mention that sometimes we lack prior knowledge of the data. Therefore, the target of our method is the data mixed with anomalies.

**Assumption 2 (bias of the number of anomalies)** *The number of anomalies is much less than the normal elements.*

The higher bias can help the model capture the statistical features to recognize the anomalies. If the numbers of abnormal and normal instances are close, our method may fail. However, if the anomalies are much more than the normal samples, we can treat the virtual normal ones as "anomalies", such as the experiment on the KDDCup99 dataset.

**Assumption 3 (independence between anomalies)** *An anomaly that occurs at an element is independent of the other anomaly occurrences in the sequence.*

Though anomalies could be relevant in real-world anomaly detection tasks, this assumption can simplify the modeling. Let $p_i$ and $p_j$ be the probabilities of anomalies occurring at elements $\boldsymbol{z}_i$

and $z_j$, respectively, then the additive probability $p_{ij} = p_i + p_j - p_i p_j$, $\bar{p}_{ij} = 1 - p_{ij}$. However, determining whether an anomaly has occurred still depends on contextual information.

**Assumption 4 (relationships among sequential elements with anomalies)** *For two elements in the sequence, if and only if at least one anomaly occurs, they have an exceptional relationship.*

We assume that only anomalies can affect the quality of a relationship between two elements in the sequence. Here, *relationship* models the degree of correlation between two elements over the sequence. Then, we need to quantify the relationship for the modeling in the following definition.

**Definition 1 (quantification of two elements having a relationship in a sequence)** *Given two elements $z_i$ and $z_j$ in a sequence, $r_{ij} \in \mathbb{R}$ quantifies the relationship between $z_i$ and $z_j$. If the two elements have an exceptional relationship, then $r_{ij} < 0$. Otherwise, $r_{ij} \geq 0$. $|r_{ij}|$ measures the strength of the relationship.*

According to Assumption 3, 4, and Definition 1, we obtain the following corollary.

**Corollary 1 (relation between sequential anomaly and relationship)** $p_{ij} = 0 \Leftrightarrow r_{ij} \geq 0$; $p_{ij} = 1 \Leftrightarrow r_{ij} < 0$.

## 3.2 PESSIMISTIC CONTRASTIVE LEARNING

We need well representations for sequential data points to apply the above assumptions to calculate the quantified relationships. As a result, we construct two pretext tasks with diverse objective functions for feature extraction and anomaly detection.

### 3.2.1 FEATURE EXTRACTION BY SEQUENTIALLY CONTRASTING

The contrastive feature extraction task mostly follows SimCLR. The encoder network in Figure 1 consists of a feature network for learning representations and a head network for projecting embeddings. Nonetheless, we perform two types of domain-agnostic feature augmentation methods for non-image and non-textual data, and modify the NT-Xent loss function for the training. The first augmentation method is adding GN $\varepsilon \sim \mathcal{N}\left(\mathbf{0}, \sigma^2 \mathbf{I}\right)$ into the embeddings (DeVries & Taylor, 2019). The noise is sampled from $\mathcal{N}\left(\mathbf{0}, \mathbf{I}\right)$ and then element-wise multiplied by $\sigma^2$, which is fit by using a full-connected layer attached to the output of the feature network. The second approach is employing DN by operating dropout with a small probability parameter in every layer of the feature network (Liang et al., 2021), where the bidirectional Kullback-Leibler divergence (biKLD) is used to regularize the augmentation. We revise Equation 1 with the regularization to the data augmentation into the following formulation:

$$\ell_{RS}(a, b) = -\frac{1}{L} \sum_{i=1}^{L} \log \frac{\exp\left(\text{sim}\left(z_i^{(a)}, z_i^{(b)}\right)/T\right)}{\sum_{n=1}^{2N} \exp\left(\text{sim}\left(z_i^{(a)}, z_i^{(n)}\right)/T\right) [n \neq a]} \tag{3}$$
$$+ \alpha D_{\text{KL}}\left(\text{softmax}\left(z_i^{(a)}\right) \| \text{softmax}\left(z_i^{(b)}\right)\right),$$

where $\alpha$ is a hyperparameter to control the regularization, and $D_{\text{KL}}$ denotes the Kullback-Leibler divergence (KLD). The regularization term in Equation 3 turns out to be the biKLD as described in (Liang et al., 2021) in the loss function for minibatches:

$$\mathcal{L}_{RS} = \frac{1}{2N} \sum_{n=1}^{N} [\ell_{RS}(2n - 1, 2n) + \ell_{RS}(2n, 2n - 1)]. \tag{4}$$

For convenience, we call this function Re-Sent[2] (regularized sequential normalized temperature-scaled cross entropy loss). In the beginning of the training, we set a certain number of epochs to solely use Equation 4 as the warm-up for better representations for the next task.

### 3.2.2 ANOMALY ESTIMATION BY CONTEXTUALLY CONTRASTING

For the sake of detecting anomalies in sequences, we leverage the contextual information given by the representative features of each sequential data point. Here, a *concept* represents a bias due to

specific sequential contexts, such as locality and periodicity. Since sequence data may have multiple concepts, we propose the MCC layer to capture the contextual information of different concepts from the embeddings of the representations learned from sequential data, as shown in Figure 1. The estimation of an anomaly probability $p_i$ is formulated by

$$\widehat{p}_i = \sigma \left( \frac{g_i}{\sqrt{m}} \right) \tag{5}$$

$$g_i = \operatorname{sgn}(G_{i,c}) \cdot \max_c (|\boldsymbol{G}_{i,:}|) \tag{6}$$

$$G_{i,c} = \frac{1}{L-2} \sum_{j=1}^{L} \boldsymbol{M}_{i,c,j} \left[ j \notin \arg\{\min, \max\}_{\mathbb{K}} (\boldsymbol{M}_{i,c,:}) \right] \tag{7}$$

$$\mathbf{M} = \tanh\left(\boldsymbol{Z} \cdot \mathbf{W}_{MCC}\right) \cdot \boldsymbol{Z}^\top, \tag{8}$$

where $\operatorname{sgn}$ indicates the sign function, $|\cdot|$ means the element-wise absolute value function, and $c$ is an index of concept. Given a sequential embedding $\boldsymbol{Z} \in \mathbb{R}^{L \times m}$, the MCC layer obtains multi-conceptual correlations $\mathbf{M} \in \mathbb{R}^{L \times C \times L}$ by performing tensor contractions on $\boldsymbol{Z}$ and its transposed copy with a learnable weight $\mathbf{W}_{MCC} \in \mathbb{R}^{m \times C \times m}$ and the $\tanh$ activation. Next, the contextual pooling (CXP) aggregates the correlations for each data point in different concepts into $\boldsymbol{G} \in \mathbb{R}^{L \times C}$. After that, the multi-conceptual pooling (MCP) selects the concept value, to which the data point is most possibly attributed, from $\boldsymbol{G}$ for each data point. Here, we adopt the robust average pooling to CXP, which calculates the mean by ignoring the largest and smallest values (Equation 7). To MCP, we use the absolute maximum pooling to select the one with the largest absolute value (Equation 6). The pooling result $\boldsymbol{g} \in \mathbb{R}^L$ is scaled by $1/\sqrt{m}$ and activated with the sigmoid function. Finally, the activation result $\widehat{\boldsymbol{p}} \in [0,1]^L$ is used to predict the anomaly probabilities. The $\tanh$ activation and the scaling can make the output values of MCC not too large and the initial values approximate to zero, i.e., the anomaly probabilities are estimated to be around 0.5 at the beginning of training.

We have modeled the anomaly probability so far. Next, we formulate the relationships in sequences by using the assumptions mentioned in Section 3.1. Due to $r_{ij} \in \mathbb{R}$ according to Definition 1, we calculate a relationship by

$$r_{ij} = \frac{\operatorname{sim}(\boldsymbol{z}_i, \boldsymbol{z}_j)}{\tau(i,j)}, \tag{9}$$

where $\tau : \mathbb{Z}^+, \mathbb{Z}^+ \to \mathbb{R}^+$ is an adaptation function. Since the sigmoid function $\sigma : \mathbb{R} \to [0,1]$ keeps the monotonicity, we can model the probability of having an exceptional relationship as

$$q_{ij} = \sigma(-r_{ij}) = 1 - \sigma(r_{ij}) = 1 - \bar{q}_{ij}. \tag{10}$$

Based on Corollary 1, the objective is to minimize the difference between the distribution of anomalies' occurrence $P$ and the distribution of data points' relationships $Q$. We could minimize the biKLD to achieve this objective. However, observing $D_{\mathrm{KL}}(P\|Q) = H(P,Q) - H(P)$ where $H(\cdot,\cdot)$ gives the cross entropy and $H(\cdot)$ calculates Shannon entropy, the second term leads the distributions into entropy maximization in the optimization. By Assumption 2, the anomaly probability is considered to be much low. Therefore, we use the following functions instead of biKLD:

$$\ell_{\beta KL}(p, q) = \frac{1}{2} \left[ D_{\beta KL}(P\|Q) + D_{\beta KL}(Q\|P) \right] \tag{11}$$

$$D_{\beta KL}(P\|Q) = \beta(p \log p + \bar{p} \log \bar{p}) - p \log q - \bar{p} \log \bar{q}, \tag{12}$$

where $\beta$ is a penalty to restrain the entropy maximization. We also clamp the probabilities with a small value to avoid $\log 0$. For the $n$-th sequence in a minibatch, we formulate the loss function as

$$\ell_{CR}(n) = \frac{1}{L-1} \sum_{i=1}^{L-1} \frac{1}{\min(s, L-i)} \sum_{j=i+1}^{\min(i+s,L)} \ell_{\beta KL} \left( p_{ij}^{(n)}, q_{ij}^{(n)} \right), \tag{13}$$

where $s$ denotes the lookahead size to handle the number of forwarding relationships to calculate. For easy reference, we term this loss function CARE (context-adaptive relative entropy loss). CARE for a minibatch of augmented embedding pairs is given by

$$\mathcal{L}_{CR} = \frac{1}{2N} \sum_{n=1}^{N} \left[ \ell_{CR}(2n-1) + \ell_{CR}(2n) \right]. \tag{14}$$

Moreover, we hope that the contextual information can reflect the representations of data points, and hence we combine Equation 4 and 14 into a global loss function for multi-task learning (Ruder, 2017). The global objective is formulated by

$$\mathfrak{L} = \lambda_{RS}\mathcal{L}_{RS} + \lambda_{CR}\mathcal{L}_{CR}, \tag{15}$$

where $\lambda_{RS}$ and $\lambda_{CR}$ are the weights for the two loss functions, respectively. When starting to optimize Equation 15, we also reduce the learning rate of the encoder network for finetuning.

## 4 EXPERIMENTS

This section reports the comparative evaluation of our proposed S[3]ADNet. We applied a simple fully connected neural network (FCNN) and a simple convolutional neural network (CNN) as the encoder models for two datasets, respectively. We show and explain the evaluation results, and give an ablation study in the following. S[3]ADNet was implemented by using Python 3.8 and PyTorch 1.8.

### 4.1 DATASETS

We evaluated S[3]ADNet on the widely used network traffic dataset KDDCup99[1], and a human activity sensing dataset HASC[2]. Table 1 summarizes the properties of the datasets.

Table 1: The properties of the datasets.

| Dataset | #instances | #features | #anomalies |
|---------|-----------|-----------|-----------|
| KDDCup99 | 494021 | 121 | 97278 |
| HASC | 39397 | 3 | 65 |

Details of the two datasets and the experimental setup for them are described in Appendix A.

### 4.2 RESULTS

We applied three S[3]ADNet variants based on the data augmentation approach used for the comparative evaluation: 1) with only GN, 2) with only DN, and 3) with both GN and DN. As S[3]ADNet can predict the anomaly probability directly, the prediction less than $0.5$ was supposed to be negative (i.e., $p_i = 0$), otherwise positive (i.e., $p_i = 1$). Additionally, we use F1-score as the primary metric for the comparison as it is better for imbalanced datasets.

Table 2 shows the result on KDDCup99, while Table 3 reports the one on HASC. On the KDD-Cup99 dataset, our method was outperformed the other state-of-the-art methods. Since the number of learnable parameters in S[3]ADNet (when $c = 8$) was only $1/15$ of the ones in MDAN, and $1/6$ of those in TS-CP[2], our proposed method is simpler and more effective.

The CPD task on the HASC dataset was challenging due to the small size and the few targets, but our proposed S[3]ADNet still achieved the highest score in the 100-margin setting and the second-highest score in the 60-margin setting. A smaller batch size was better for this task. However, the training was unstable, and the F1-scores oscillated in both two tasks. We will continue the discussion on the problems in the next section.

### 4.3 ABLATION STUDY

We report the best scores for each hyperparameter settings in the following. The default values of the hyperparameter are shown in Table 4. The models were trained five times in each setting.

---

[1]http://kdd.ics.uci.edu
[2]http://hasc.jp

Table 2: The results on the KDDCup99 dataset. The **bold** score denotes the highest one in the column. We report the best scores for $S^3$ADNet.

| Model | Precision | Recall | F1-score |
|---|---|---|---|
| OC-SVM (Schölkopf et al., 1999) | 0.7457 | 0.8523 | 0.7954 |
| IF (Liu et al., 2008) | 0.9216 | 0.9373 | 0.9294 |
| DSEBM-r (Zhai et al., 2016) | 0.8521 | 0.6472 | 0.7328 |
| DSEBM-e (Zhai et al., 2016) | 0.8619 | 0.6446 | 0.7399 |
| DAGMM (Zong et al., 2018) | 0.9297 | 0.9442 | 0.9369 |
| AnoGAN (Schlegl et al., 2017) | 0.8786 | 0.8297 | 0.8865 |
| ALAD (Zenati et al., 2018) | 0.9427 | 0.9577 | 0.9501 |
| MDAN (Hou et al., 2020) | 0.9472 | 0.9623 | 0.9547 |
| $S^3$ADNet (GN) | 0.9337 | 0.9633 | 0.9482 |
| $S^3$ADNet (DN) | 0.9389 | 0.9829 | 0.9604 |
| $S^3$ADNet (GN+DN) | 0.9453 | 0.9779 | **0.9613** |

Table 3: The results on the HASC dataset. The margin is the maximum allowed detection error (the number of data points from a ground truth to an estimated change point) that within this range, the estimation is still considered a true positive. In the columns, the **bold** and underlined scores denote the highest and second-highest scores, respectively. We report the best F1-score of each method.

| Margin | 60 | 100 | 200 |
|---|---|---|---|
| Model | | F1-score | |
| FLOSS (Gharghabi et al., 2019) | 0.3088 | 0.3913 | 0.543 |
| aHSIC (Yamada et al., 2013) | 0.2308 | 0.3134 | 0.4167 |
| RuLSIF (Liu et al., 2013) | 0.3433 | 0.4999 | 0.4999 |
| ESPRESSO (Deldari et al., 2020) | 0.2879 | 0.4233 | **0.6933** |
| KL-CPD (Chang et al., 2019) | **0.4785** | 0.4726 | 0.4669 |
| TS-CP$^2$ (Deldari et al., 2021) | 0.40 | 0.4375 | 0.6316 |
| $S^3$ADNet (GN) | 0.4783 | **0.5389** | 0.6225 |
| $S^3$ADNet (DN) | 0.4471 | 0.5316 | 0.6225 |
| $S^3$ADNet (GN+DN) | 0.4457 | 0.5304 | 0.6214 |

Table 4: Default hyperparameters for the ablation study. The meanings of the symbols are explained in Appendix A.

| Dataset | $L$ | $N$ | $C$ | $r$ | $T$ | $k$ | $\alpha$ | $\beta$ | $\lambda_{CR}$ | #warm-ups |
|---|---|---|---|---|---|---|---|---|---|---|
| **KDDCup99** | 8 | 256 | 8 | 0.5 | 0.05 | 0.25 | 0.1 | 0.1 | 5 | 5 |
| **HASC** | 4 | 8 | 8 | 0.5 | 0.05 | 0.25 | 1 | 1 | 3 | 10 |

Table 5: Best F1-scores obtained with different numbers of concepts on the two datasets.

| Dataset | KDDCup99 | | | | HASC ($w$=100) | | | |
|---|---|---|---|---|---|---|---|---|
| $C$ | 1 | 4 | 8 | 12 | 1 | 4 | 8 | 12 |
| F1-score | 0.7576 | 0.9486 | 0.9242 | 0.9431 | 0.3649 | 0.4762 | 0.4792 | 0.5056 |

### 4.3.1 IMPACTS OF THE NUMBER OF CONCEPTS

The results shown in Table 5 are obtained by selecting $c \in \{1, 4, 8, 12\}$ to train $S^3$ADNet with DN on KDDCup99 and HASC. We found that more than one concept could help improve the performance. Nevertheless, because the weight **M** is dense, a large $c$ increases the number of parameters and may affect the convergence of training.

Table 6: Best F1-scores obtained with different temperatures and adaptation functions on HASC ($w$=100).

| $T$ | $k$ | constant | log | root | exp |
|---|---|---|---|---|---|
| 0.05 | 0.1 | 0.41 | 0.4162 | 0.4471 | 0.4158 |
| | 0.25 | 0.4513 | 0.4815 | 0.4340 | 0.4815 |
| | 0.5 | 0.4815 | 0.3776 | 0.3983 | 0.4052 |
| | 1 | 0.4324 | 0.5202 | 0.3882 | 0.4571 |
| 0.1 | 0.1 | 0.3902 | 0.4198 | 0.4590 | 0.4605 |
| | 0.25 | 0.4444 | 0.3636 | 0.4123 | 0.4416 |
| | 0.5 | 0.4444 | 0.3907 | 0.4342 | 0.4063 |
| | 1 | 0.4835 | 0.4105 | 0.3908 | 0.4365 |

### 4.3.2 IMPACTS OF ADAPTATION FUNCTIONS

Since the adaptation function for the KDDCup99 dataset was constant, we used the results on HASC to analyze the effect of adaptation functions. In this case, we selected $T \in \{0.05, 0.1\}$ and $k \in \{0.1, 0.25, 0.5, 1\}$ with different adaptation functions. We found that a smaller temperature was better for variant adaptation functions.

## 5 DISCUSSION

Notwithstanding the outperformance of S$^3$ADNet in the evaluation, there are still some limitations in our method. Firstly, similar to the other deep learning methods, our method requires many hyperparameters to set up the model. Although several hyperparameter optimization algorithms are available to us (Yu & Zhu, 2020), they need validation sets and specific metrics for the trials. While our method is unsupervised/self-supervised, it could not be swift enough to adapt to an entirely new dataset having no validation set. A potentially viable solution is using the ensemble learning technique to combine the results from several models with low prediction losses.

Nevertheless, there was the second problem. We observed that the training of S$^3$ADNet could be unstable like the GAN-based methods. This problem results in the difficulty of selecting well enough models by the prediction losses. One possible reason is that the value instability from KLD. We have attempted to use the Jensen–Shannon divergence instead of KLD, but the performance was even worse. It is the future work to seek a better measure. Another cause could be in the modeling of relationships. Before the context learning stage, the encoder has been trained to make the identical instances have the representations as similar as possible but not concerning the contextual relationships. After that, however, the identical relationship could be broken by optimizing Equation 14. It seems reasonable to attempt some tricks for training stabilization, such as stop-gradient (Chen & He, 2021). One more reason could be that the degrees of freedom of the MCC weight were too high due to the density. It may be helpful to employ sparsification or factorization to the weight.

Thirdly, there are still potential options to improve the performance. We proposed a method based on the density assumption so far. Since the hybrid method ESPRESSO acquired the top score in the 200-margin setting of the CPD task on HASC, it could be wise to take advantage of other principles in our model. Furthermore, though we have applied a simple FCNN and a simple CNN as the encoder networks in the experiment, we have not investigated the selection of the hyperparameters of encoders in depth. Also, we found that they were still hard to extract representative features from complex multivariate time-series data, such as SMAP and MSL (Hundman et al., 2018), SMD (Su et al., 2019). It needs more studies on selecting encoder networks for those data, such as leveraging RNN to learn representations of short sequences.

## 6 CONCLUSION

This paper proposed a flexible and straightforward framework S$^3$ADNet, for detecting anomalies on sequential data by using contrastive learning under the pessimistic assumptions. The future work is to stabilize the training and attempt to leverage other aspects to improve the performance.

ACKNOWLEDGMENTS

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

## A  DETAILS OF EXPERIMENT SETUP

### A.1  DETAILS OF DATASETS

- **KDDCup99** (Stolfo et al., 2000). In the experiment, we used the KDDCup99 10% dataset. As mentioned in Section 3.1, the number of records labeled as "normal" in the dataset is highly less than that of intrusions, so that we regarded the instances labeled as attacks as positive samples for the SAD task. For each variable, the discrete ones were one-hot encoded, while the contiguous ones were standardized by removing the median and scaling to unit variance according to the first and third quartile values. We split the dataset into two random halves, and employed the first half of data for training and the rest for testing in each epoch.

- **HASC** (Kawaguchi et al., 2011). We utilized the same HASC subset used by KL-CPD and TS-CP[2], having only the 3-axis accelerometer logs. The target of the task is to detect the human activity pattern changes. As following the setting in (Deldari et al., 2021), we set three chunk sizes $w \in \{60, 100, 200\}$ to vary the subset to three chunked versions and used the whole subset for both the training and testing in the evaluation. The values of each axis were standardized by removing the mean and scaling.

For each dataset, the samples in a minibatch were generated in a sliding window fashion, where the window size was the sequence length $L$, and the stride was $1$.

### A.2  MODEL SETUP

According to the properties of the two datasets, we applied diverse but simple encoder networks for them. The details of the architectures and the hyperparameter settings are described as follows.

- KDDCup99. Due to the sparsity of each data point $\boldsymbol{x} \in \mathbb{R}^{121}$, we employed a 2-layer FCNN for the feature network and a 1-layer FCNN for the head network. In the feature network, the output size of each layer was $32$. The embedding size of the head network was $m = 8$. For the hyperparameter optimization, the sequence length $L \in \{4, 8, 16\}$, the batch size $N = 256$, and the warm-up was up to $\{5, 10, 20\}$ epochs. Since the data points were shuffled, we set the adaptation function to $\tau(i, j) = k$ (*constant*).

- HASC. Depending on the chunk size $w$, each data point was a dense matrix $\boldsymbol{X} \in \mathbb{R}^{3 \times w}$. This time, we applied a 2-layer 1D CNN for the feature network but a 1-layer FCNN for the head network. For each convolution layer, the output channel size was $32$, the kernel size was selected from $\{3, 5\}$, and each input's head and tail were padded with a zero. A maximum pooling and an average pooling were adapted to each output channel from the second CNN layer to extract the concatenated representative features with the size $64$. Then, the representations were projected to embeddings with the size $m = 16$. For the hyperparameter selection, the sequence length $L \in \{4, 8\}$, the batch size $N \in \{4, 8\}$, and the number of warm-up epochs were chosen from $\{20, 30\}$. The adaptation function was selected from *constant*, $\tau(i, j) = k \ln(j - i + 1)$ (*log*), $\tau(i, j) = k\sqrt{j - i}$ (*root*), and $\tau(i, j) = 1.1^{j-i}k$ (*exp*), making the relationships with further data points weaker.

The following settings were common for both datasets. The activation was LeakyReLU (Maas et al., 2013) with a negative slope of $0.2$ in both of the feature networks. For the data augmentation using DN, the dropout probability was set to $0.1$. The objective functions were optimized by applying RMSProp (Graves, 2013) with the learning rate $0.1$, the momentum $0.9$, the smoothing parameter

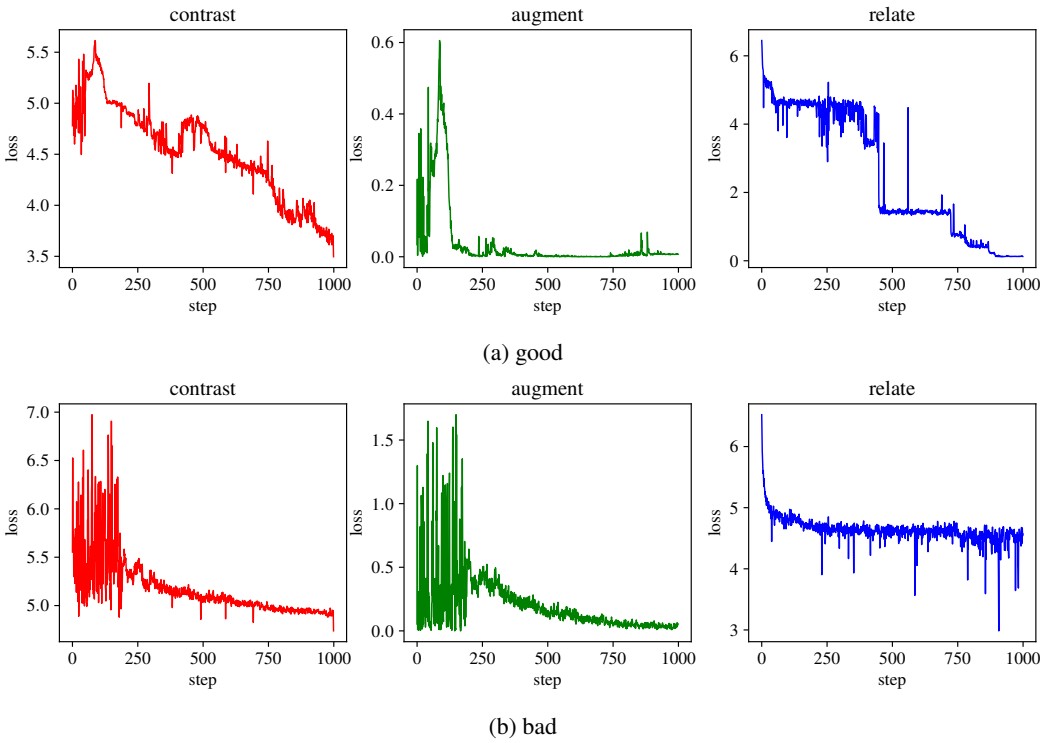

Figure 2: Two examples of loss patterns on KDDCup99. Losses of *contrast* and *augment* were obtained from the first and second terms in Equation 3, respectively, and ones of *relate* were obtained from Equation 13. Figure 2a led to a good F1-score, while Figure 2b did not.

0.99, and the weight decay $1.0 \times 10^{-4}$. Since KLD could approach infinity, the gradients were clipped in the range $[-10, 10]$. After the warm-up epochs, the learning rate for the encoder network was reduced to $1.0 \times 10^{-4}$ for the finetuning. For the multi-task learning, we set $\lambda_{RS} = 1$ and $\lambda_{CR} \in \{3, 5\}$. Every training trial was up to 100 epochs. The lookahead size was given by $s = \lfloor rL \rfloor$, where $r \in \{0.25, 0.5, 0.75\}$. Others were, the number of concepts $C \in \{1, 4, 8, 12\}$, the regularization weight $\alpha \in \{0.1, 0.5, 1\}$, the entropy penalty $\beta \in \{0.1, 0.5, 1\}$, and the temperature $T \in \{0.05, 0.1\}$. Furthermore, the coefficients $k$ in adaptation functions were selected from $\{0.1, 0.25, 0.5, 1\}$,

## B   EXAMPLES OF TRAINING LOSS PATTERNS

Figure 2 shows two examples of the training loss patterns on KDDCup99. The changes of *contrast* losses and *augment* losses were intense in warm-up epochs. If the hyperparameter setting is selected appropriately, there could be significant and sustainable drops in *relate* losses, as shown in Figure 2a. The instability in Figure 2b may be attributed to KLD.

