# OpenReview forum: "S$^3$ADNet: Sequential Anomaly Detection with Pessimistic Contrastive Learning"
_ICLR.cc/2022/Conference — ICLR 2022 Submitted_

### Official Review · Reviewer_kP8v · 2021-11-02

**Correctness:** 3
**Technical Novelty And Significance:** 3
**Empirical Novelty And Significance:** 2
**Recommendation:** 5
**Confidence:** 4

**Main Review:**

### Strength
1) The paper is mostly well written.
2) The idea is interesting.

### Weakness
1) The empirical analysis lacks depth.
2) Many recent SOTA methods are missing from the analysis.
3) Many recent anomaly detection dataset such as SMAP [a], MSL [a], SWAT, and SMD [b] are missing from the evaluation.
4) The literature review include most of the recent methods but some recent papers on anomaly detection are missing. For example OmniAnomaly [b], THOC [c], and other [d],[e],[f] etc.
5) Missing a proper ablation study of the proposed method.

### References
[a] Hundman, Kyle, et al. "Detecting spacecraft anomalies using lstms and nonparametric dynamic thresholding." Proceedings of the 24th ACM SIGKDD international conference on knowledge discovery & data mining. 2018.

[b] Su, Ya, et al. "Robust anomaly detection for multivariate time series through stochastic recurrent neural network." Proceedings of the 25th ACM SIGKDD International Conference on Knowledge Discovery & Data Mining. 2019.

[c] Shen, Lifeng, Zhuocong Li, and James Kwok. "Timeseries anomaly detection using temporal hierarchical one-class network." Advances in Neural Information Processing Systems 33 (2020): 13016-13026.

[d] Tariq, Shahroz, et al. "Detecting anomalies in space using multivariate convolutional LSTM with mixtures of probabilistic PCA." Proceedings of the 25th ACM SIGKDD International Conference on Knowledge Discovery & Data Mining. 2019.

[e] Yairi, Takehisa, et al. "A data-driven health monitoring method for satellite housekeeping data based on probabilistic clustering and dimensionality reduction." IEEE Transactions on Aerospace and Electronic Systems 53.3 (2017): 1384-1401.

[f] Shin, Youjin, et al. "ITAD: Integrative Tensor-based Anomaly Detection System for Reducing False Positives of Satellite Systems." Proceedings of the 29th ACM International Conference on Information & Knowledge Management. 2020.

**Summary Of The Paper:**

The author proposes a self-supervised sequential anomaly detection network that optimizes a context-adaptive objective using feature argumentation and contextual information. The proposed method outperforms baselines.

**Summary Of The Review:**

The idea is interesting but the paper lacks depth in empirical analysis.

---

> ### Author Response · Authors · 2021-11-22
> **Response to Reviewer kP8v**
>
> Thank you for the comments as well as pointing out the new references. We are glad you found the paper very interesting and well-written.
>
> - **The empirical analysis lacks depth; missing a proper ablation study of the proposed method.**
>
> We have added an ablation study and examples for training loss patterns of individual loss terms into the latest submission.
>
> - **Many recent SOTA methods are missing from the analysis; many recent anomaly detection dataset such as SMAP [a], MSL [a], SWAT, and SMD [b] are missing from the evaluation.**
>
> In this paper, we focus on proposing the framework of pessimistic contrastive learning, so that we evaluate the method on KDDCup99 and HASC by applying simple networks as the base models. However, we tested our method on SMAP and MSL roughly and found that the performance was not good by using those simple networks. It needs more future work to select appropriate base models for these datasets. We added the content about this in the Discussion section.
>
> - **The literature review include most of the recent methods but some recent papers on anomaly detection are missing.**
>
> We have added those references into the Introduction and Related Work sections.

---

> > ### Comment · Reviewer_kP8v · 2021-11-29
> > **Thank you**
> >
> > I would like to thank the authors for their honest response. I believe that the state of empirical analysis at the moment is insufficient to meet ICLR criteria. Therefore, I will stay with my original score.

---

### Official Review · Reviewer_6q4k · 2021-11-03

**Correctness:** 3
**Technical Novelty And Significance:** 2
**Empirical Novelty And Significance:** 2
**Recommendation:** 3
**Confidence:** 4

**Main Review:**

·	Strength:

o	The paper is easy to follow.

o	The empirical result on two datasets outperforms most of the baselines.

·	Weakness:

o	The acknowledgements in the paper implicitly reveal the identity of authors, which may against the submission policy.

o	The novelty of this work is limited. In this work, only multi-conceptual pooling layer is the newly developed components.

o	The two feature augmentation methods seems arbitrary as there are no justifications for the reason of adopting the two methods.
Providing more justifications may alleviate the concern.

o	The key assumption underlying the model is Assumption 3, which arbitrarily assume that there is no causal relations between anomalies, which may not be applicable to various real-world applications such as fault detection. More justification on this assumption may alleviate the concern.

o	The anomalies in the adopted dataset seems not following the Assumption 3. For example, the human activity patterns in HASC dataset are not completely independent to each other, which makes it surprising that the model works on this case. More explanation on this part may be helpful to clarify the model capability.

o	The selection of adaptation function seems arbitrary in the experiment, maybe more explanations will help readers to understand the function selection and development.

o	Since the main contribution of this work is multi-conceptual pooling layer, I would expect more empirical analysis on this components. For example, how does the “C” affecting the outlier detection and what is the selection criteria? More explorations on this will certainly help readers to understand about how to properly use the model.


**Summary Of The Paper:**

·	The author propose a contrastive learning framework for sequential outlier detection.

·	The author extract sequential feature representation for individual data points by applying dropout and Gaussian noise to SimCLR.

·	The author propose a contextual contrastive loss via developing a multi-conceptual pooling layer to minimize the distance between the probability distribution of anomalies and whole datasets.


**Summary Of The Review:**

·	To summarize, this paper proposed a pessimistic contrastive learning framework for sequential outlier detection. The proposed method is technically sound. However, the novelty is limited on multi-conceptual pooling and no further empirical exploration on this component is provided. Also, many selections for the model seems arbitrary, which needs more justifications to support the motivation; and the dataset selection seems conflict to the model assumption. The most important part is the author includes the acknowledgements in the draft, which implicitly reveal the author identity and is against the submission policy. Therefore, I recommend rejection to this work.

---

> ### Author Response · Authors · 2021-11-22
> **Response to Reviewer 6q4k**
>
> We would like to thank the reviewer for the comments and suggestions. We have addressed the reviewer's comments one by one below.
>
> - **The acknowledgements in the paper implicitly reveal the identity of authors, which may against the submission policy.**
>
> The submission policy only writes "reviewers cannot see author names when conducting reviews, and authors cannot see reviewer names", and we cannot be immediately and individually identified via our paper. Therefore, we consider that our paper is not against the policy. However, we have removed this part in the latest submission to avoid potential problems the reviewer pointed out.
>
> - **The novelty of this work is limited. In this work, only multi-conceptual pooling layer is the newly developed components.**
>
> Besides the multi-conceptual context layer, we propose a new loss function CARE based on modeling the relationship between sequential elements and introducing a penalty to control the entropy maximization for imbalanced data into the KL divergence.
>
> - **The two feature augmentation methods seem arbitrary as there are no justifications for the reason of adopting the two methods.**
>
> We have added the justification for the use of the two augmentation methods in Section 3.2.1.
>
> - **The key assumption underlying the model is Assumption 3, which arbitrarily assume that there is no causal relations between anomalies, which may not be applicable to various real-world applications such as fault detection.**
>
> This assumption is for simplifying the modeling. Also, we think that an additional module for capturing the relationship between anomalies is meaningful but leave it for future work.
>
> - **The anomalies in the adopted dataset seems not following the Assumption 3.**
>
> We have removed the original Assumption 3 and attempted to clarify the descriptions in Section 3.1.
>
> - **The selection of adaptation function seems arbitrary in the experiment; how does the “C” affecting the outlier detection and what is the selection criteria?**
>
> We have added an ablation study which may alleviate the above concerns.

---

> > ### Comment · Reviewer_6q4k · 2021-11-29
> > **Reply**
> >
> > Thank you for the detailed reply to my concerns. Also thank you for the additional experiments on the ablation study.  The authors are encouraged to improve the paper with an improved discussion of more recent work and polish the paper from incorporating reviewers' comments, and resubmit to an appropriate venue. I will stay with my original score.

---

### Official Review · Reviewer_Gpfn · 2021-11-03

**Correctness:** 2
**Technical Novelty And Significance:** 2
**Empirical Novelty And Significance:** 1
**Recommendation:** 1
**Confidence:** 4

**Main Review:**

This paper has several significant issues with the primary being imprecise and unclear technical description of their method:

* __Lack of Clarity:__ Many parts of this paper are technically imprecise, incorrect, or too implicit. This issue is severe enough that I do not think one could implement the method from the description in the paper. This alone is enough for a paper to receive a strong reject. I will give some examples here:
  * __S3.1 first paragraph:__ the definition of $\mathbf{z}_i$ is inconsistent with the rest of the paper, here it represents data points, later it represents an embedding. On p. 6 it is introduced as an embedding however it is used in situations where it doesn't make sense before that, specifically (6).
  * __Assumption 3:__ This assumption needs to be made more concrete, in particular independence between anomalies conditioned on what information? Total independence doesn't really make sense since this work is specifically working on _time series_ anomaly detection means there is some sort of dependence between different time points. Maybe we are assuming some model where the anomalous points are chosen independently a-priori and then theres a model that generates the time series conditioned on the location of the anomalies?
  * __p.4__ "Unlike one-class learning, we assume that there are anomalies in the training data." This is incorrect. For the one class support vector machine, for example, there are slack variables that explicitly allow for points in the training dataset to fall in the anomalous region
  * __Assumption 4:__ "For two data points in the sequence, if and only if no anomaly occurs, they have a normal or no relationship." This sentence is inscrutable. No anomaly anywhere? No anomalies at the points? At one of the points? What is a "relationship"? Is "normal" and "no relationship the same thing"? Does "normal" have something to do with the normal distribution?  This kind of language is unacceptable.
  *__Assumption 5:__ Similar issues as the last point.
  *__Definiton 1:__ See above.
  *__(3):__ What is the motivation for the KL term?  It seems very ad hoc.
  *__"Tensor product"__ This is being used incorrectly. For two tensors $S\in \mathbb{R}^{d_1\times d_2\times \cdots \times d_m}$ and $T \in \mathbb{R}^{k_1\times k_2\times \cdots \times k_n}$ the tensor product $S\otimes T$ will lie in $\mathbb{R}^{d_1\times d_2\times \cdots \times d_m \times k_1\times k_2\times \cdots \times k_n}$ and be rank one in $\mathbb{R}^{d_1\times d_2\times \cdots \times d_m} \times \mathbb{R}^{k_1\times k_2\times \cdots \times k_n}$. This is not how you are utilizing it in (8) for example where you are performing tensor multiplication along the first mode of $Z$ and $W$, for example. This needs to be introduced and explained. Also note that with this sort of notation transposes are not necessary.
  *__"tensor weight"__ $\mathbf{W}_{MCC}$ (4 lines below (8)): How is this selected? Is it learned? This is the only occurrence of this in the whole paper from what I can tell and how it is selected, learned, or what it means is never explained.
  * __"concept":__ This needs some introduction and some intuition.
  * There are many more instances of this issue, but being exhaustive would take far too long
 *__Insufficient Experimental Results:__ This paper needs a very comprehensive experimental evaluation since the theoretical and conceptual contributions are small. More than 2 datasets and more competitors, including better shallow baselines especially for the 3 dimensional dataset. Would be interesting to see _data depth_ based methods as well.

**Summary Of The Paper:**

In this paper the authors present a method for deep anomaly detection. They being by stating a few assumptions their method is based on. Then then develop two loss terms. The first loss term is based on contrastive learning with an additional term. Their second term is used for determining the anomaly score and is based on how well each "data point" fits to concept.

They validate their method on a couple of datasets and their method works pretty well.

**Summary Of The Review:**

I recommend the paper be rejected primarily due to its exposition not being clear and technical aspects of the paper being unclear or wrong.

---

> ### Author Response · Authors · 2021-11-22
> **Response to Reviewer Gpfn**
>
> We would like to thank the reviewer for the comments and suggestions. However, we believe that there are significant misunderstandings in the reviewer’s opinions on our paper. Please see our response below.
>
> - **The definition of zi is inconsistent with the rest of the paper, here it represents data points, later it represents an embedding.**
>
> We considered the "data point" here as an abstract concept that means an element in a sequence, i.e., for an input sample in terms of raw data, or for an embedding in terms of sequential embeddings. Because we would use it in modeling the sequential embeddings, we used $z_i$ instead of $x_i$ to preserve the consistency to the following application of the assumptions. However, we have changed all the "data point"s to "element"s in Section 3.1 for clarity.
>
> - **Total independence doesn't really make sense since this work is specifically working on time series anomaly detection means there is some sort of dependence between different time points.**
>
> Firstly, our method is not specifical for time series data, but general for ones who can be constructed as sequences. For time series anomalies, there can indeed be some sort of dependence between them. However, we take Assumption 3 for the simplification of the model. We also think that an additional module for capturing the relationship between anomalies is meaningful but leave it for future work.
>
> - **"Unlike one-class learning, we assume that there are anomalies in the training data.\" This is incorrect.**
>
> We had intended to point out the naive idea about one-class. We modified the description here.
>
> - **About Assumption 4, 5, and Definiton 1**
>
> We have removed the original Assumption 3 to avoid confusion and added a description about "relationship" before Definition 1.
>
> - **(3): What is the motivation for the KL term?**
>
> It is derived from (Liang et al., 2021) we mentioned. However, we added a description of the use of the KL term.
>
> -  **"Tensor product" This is being used incorrectly.**
>
> Though we used $\otimes$ for the operation unconventionally, we mentioned that it was for tensor contractions (tensor dot product) but not for tensor product. However, we have changed the symbol to $\boldsymbol{\cdot}$ now.
>
> - **WMCC (4 lines below (8)): How is this selected? Is it learned?**
>
> It is learned from data. We have put "learnable" to modify it.
>
> - **"concept": This needs some introduction and some intuition.**
>
> We have added a description of it in Section 3.2.2 and analysis results on it in the ablation study.
>
> - **Insufficient Experimental Results**
> We have added an ablation study and examples of training loss patterns in the latest submission.

---

### Official Review · Reviewer_KQ2g · 2021-11-04

**Correctness:** 3
**Technical Novelty And Significance:** 3
**Empirical Novelty And Significance:** 3
**Recommendation:** 5
**Confidence:** 3

**Main Review:**

Strong points:
+ All of the assumptions and the subsequent corollary are clearly presented.
+ The design of the novel loss function based on a pessimistic policy has been discussed in detail.
+ Many state-of-the-art methods are used as benchmark studies and are compared with the proposed method.
Thorough discussions are provided on observations related to unstable training and possible ways to mitigate it in future work.

Weak points:
- I’m not sure how well Assumption 3 applies to the real-world sequential data. If an anomaly occurs at a point in time, it may typically take some time to recover and result in anomalies present in adjacent data points. If that’s the case, it will make Assumption 3 that assumes independence between anomalies at stake.
- For the real-world dataset used in the experiment, such as KDDCup99, is the occurrence of an anomaly still independent of the other anomaly occurrences in the sequence? The authors have explicitly stated that the proposed method requires sequential information, so the dataset was not split up randomly.
- It is mentioned in Section 4.3 that “As S$^3$ADNet can predict the anomaly probability directly, the prediction less than 0:5 was supposed to be negative, otherwise positive.” This seems to be an extension of Corollary 1 and needs more explanations or proof.
- The last sentence of Section 1 claims that “Our proposed S$^3$ADNet obtained competitive F1-score results to those methods with a simpler network architecture.” However, no comparative results are provided that justify the proposed method is of simpler network architecture or has a fewer number of parameters than the state-of-the-art methods listed in Tables 2 and 3. To justify such a claim, the authors are advised to present a figure similar to Fig. 1 of the original SimCLR paper.
- More details should be provided on the number of hyperparameters to set up the model, an ablation study is suggested to show the robustness of the proposed method using suboptimal hyperparameters.


**Summary Of The Paper:**

Inspired by SimCLR, this paper proposes a simple neural network framework for detecting anomalies on sequential data. By first performing feature augmentation and then estimating anomaly probabilities from the sequential data points by optimizing the context-adaptive objective, it is shown that the proposed method - S$^3$ADNet, outperforms many state-of-the-art approaches on two benchmark datasets in terms of the F1-score.

**Summary Of The Review:**

It is indeed interesting and of practical value to distinguish any anomaly condition using sequential data with simpler network architecture. However, there are some places where this paper needs to clarify and provide more details to make it technically sound.

---

> ### Author Response · Authors · 2021-11-22
> **Response to Reviewer KQ2g**
>
> Thank the reviewer for the thoughtful feedback. We answer the reviewers concerns in the following:
>
> - **If an anomaly occurs at a point in time, it may typically take some time to recover and result in anomalies present in adjacent data points.**
>
> Yes, this case may occur in the real world. We introduced this assumption to simplify the modeling. However, we also think that an additional module for capturing the relationship between anomalies could help model the case. We leave it for future work.
>
> - **For the real-world dataset used in the experiment, such as KDDCup99, is the occurrence of an anomaly still independent of the other anomaly occurrences in the sequence?**
>
> In the latest submission, we have improved the experiment on KDDCup99 by randomly splitting the dataset.
>
> - **This seems to be an extension of Corollary 1 and needs more explanations or proof.**
>
> We have modified the symbol for predictive probabilities in Equation 3, added supplementary descriptions in Section 4.1.
>
> - **No comparative results are provided that justify the proposed method is of simpler network architecture or has a fewer number of parameters than the state-of-the-art methods**
>
> We have added comparative results to MDAN and TS-CP$^2$ upon the number of parameters in Section 4.2.
>
> - **An ablation study is suggested to show the robustness of the proposed method using suboptimal hyperparameters.**
>
> We have added an ablation study in Section 4.2.

---

### Official Review · Reviewer_2onX · 2021-11-05

**Correctness:** 3
**Technical Novelty And Significance:** 4
**Empirical Novelty And Significance:** 2
**Recommendation:** 5
**Confidence:** 4

**Main Review:**

[Strengths]
As the title indicates "Sequential anomaly detection", the main idea strongly involves the sequential properties of data.
The loss design of minimizing KL-divergence between relationship probability and anomaly probability is interesting and intuitive.
The writing is easy to follow and clear.
Assumptions are well addressed.
The discussion section is very helpful to readers addressing existing limitations and suggesting future directions.
The proposed method shows strong performance against competing methods.


[weaknesses + feedback]
- Although I like the idea of minimizing the distribution gap between anomaly probability and sequential relations, the model is complex with many hyperparameters (the authors already mentioned in the discussion section). Also, the choice of adaptation function, augmentation methods are adding more options for the model. In short, the design space is too broad, and hard to search for the optimal model. It would be great if the authors could suggest a way to narrow down this design space without using labels.  For example, by analyzing loss patterns of individual loss terms. Or the use of a small validation set can be an option.

- Because of broad design space (and hyperparameters), the authors tested variants of the model and reported the mean performance.
I suggest providing ablation studies of these hyperparameters to figure out which one is sensitive and which one is not. Without an ablation study, it is hard to measure which part of the model design plays a critical role in the performance. And I suggest providing a standard deviation of these models' performances to show whether the performance gap between variants is high or low.

[feedback]
As suggested in the discussion section, I agree that the ensemble approach can be a good solution.
The paper would be stronger if the authors could reveal sensitive hyperparameters to reduce the critical hyperparameters for tuning.
Then build up an ensemble based on the models with varying sensitive hyperparameters can be one good approach to solve design complexity.

**Summary Of The Paper:**

The paper tackles the anomaly detection of time series data. The main idea is to exploit both the sequential relation between data and the normality of the data itself. The model is learned with two stages. First, the model is learned with the sequential contrastive loss to learn data representations. Then the model is learned by minimizing the distribution between normality and exceptional relationship probability.
The learning process is a self-supervised way without any supervision. The model is validated with KDDCup99 and HASC datasets.


**Summary Of The Review:**

The idea is interesting and I like it.
I highly acknowledge the discussion section.
However, the model design choices are too broad (hyperparameters, augmentation, and adaptation functions) and necessary ablation studies are missing.
If analysis on important design factors can be provided, the paper would be more thorough.

---

> ### Author Response · Authors · 2021-11-22
> **Response to Reviewer 2onX**
>
> Thank the reviewer for the helpful review and positive opinion. We are glad the reviewer found the paper very interesting.
>
> In the latest submission, we have added an ablation study in Section 4.3, and given examples of training loss patterns of individual loss terms for reference. We hope they can alleviate the reviewer’s concerns.

---

### Author Response · Authors · 2021-11-22
**To all reviewers**

We thank all reviewers for their constructive feedback. The summary of the changes is in the following.

- Removed the original Assumption 3 and attempted to clarify the descriptions in Section 3.1.
- Added an ablation study in Section 4.3.
- Improved the experiment on KDDCup99 by randomly splitting the dataset, updated some evaluation results in Table 2 and 3, and compared the model size to two state-of-the-art models in Section 4.2.
- Added discussion about complex time series data.
- Moved details of the datasets and the experiment setup to Appendix A.
- Added examples of loss patterns for reference in Appendix B.

---

### Decision · Program_Chairs · 2022-01-20

**Decision:**

Reject

**Comment:**

The work proposes a simple neural network framework for detecting anomalies on sequential data. The manuscript is quite rough. The paper needs significant editing. The authors should take the reviewers' recommendations to heart and make deeper changes to the paper.